

# The potential of global re-analysis datasets in identifying flood events in Southern Africa

Gaby J. Gründemann[1], Micha Werner[1,2], Ted I.E. Veldkamp[3]

[1]IHE Delft, 2601 DA, Delft, the Netherlands
[2]Deltares, 2600 MH, Delft, the Netherlands
[3]Institute for Environmental Studies, Vrije Universiteit Amsterdam, 1081 HV, Amsterdam, the Netherlands

*Correspondence to*: Gaby J. Gründemann (gabygrundemann@gmail.com)

**Abstract.** Sufficient and accurate hydro-meteorological data are essential to manage water resources. Recently developed global re-analysis datasets have significant potential in providing these data, especially in regions such as Southern Africa that
are both vulnerable and data poor. These global re-analysis datasets have, however, not yet been exhaustively validated and it is thus unclear to what extent these are able to adequately capture the climatic variability of water resources, in particular for extreme events such as floods. This article critically assesses the potential of a recently developed global Water Resource Re-analysis (WRR) dataset developed in the EU FP7 eartH2Observe project for identifying floods, focussing on the occurrence of floods in the Limpopo River basin in Southern Africa. The discharge outputs of seven global models and ensemble mean
of those models as available in the WRR dataset are analysed and compared against two benchmarks of flood events in the Limpopo River basin. The first benchmark is based on observations from the available stations, while the second is developed based on flood events that have led to damages as reported in global databases of damaging flood events. Results show that while the WRR dataset provides useful data for detecting the occurrence of flood events in the Limpopo River basin, variation exists amongst the global models regarding their capability to identify the magnitude of those events. The study also reveals
that the models are better able to capture flood events at stations with a large upstream catchment area. Improved performance for most models is found for the 0.25 degrees resolution global model, when compared to the lower resolution 0.5 degrees models, thus underlining the added value of increased resolution global models. The skill of the global hydrological models in identifying the severity of flood events in poorly gauged basins such as the Limpopo can be used to estimate the impacts of those events using the benchmark of reported damaging flood events developed at the basin level, though could be improved
if further detail on location and impacts are included in disaster databases. Large-scale models such as those included in the WRR dataset are used by both global and continental forecasting systems, and this study sheds light on the potential these have in providing information useful for local scale flood risk management. In conclusion, this study offers valuable insights in the applicability of global re-analysis data for identifying impacting flood events in data sparse regions.





## 1 Introduction

Floods are among the most common and destructive natural hazards globally (Jongman et al., 2015). Approximately 90% of disasters worldwide in the last decades were caused by weather-related events. Among them, floods are the most frequent, and affected 2.3 billion people between 1995 and 2015 (UNISDR & CRED, 2015). It is generally acknowledged that due to

projected climate and socio-economic changes, extreme events such as floods may further increase in frequency, magnitude and intensity (IPCC, 2012, 2014, UNISDR, 2015, 2016). In order to minimise the negative effects of floods, disaster risk reduction is increasingly important (Trigg et al., 2016). The urgency of mitigating flood risks is also recognised by international agreements, such as the Sendai Framework for Disaster Risk Reduction (UNISDR, 2015), which underlines the understanding of disaster risk including the hazard characteristics as a first priority. Developing adequate knowledge of past flood events is

essential in order to sufficiently address this global problem (Dottori et al., 2016; Spaliviero et al., 2011) and to further reduce the consequences of future disastrous events.

Accurate data is key to developing a reliable representation of floods. While hydro-meteorological data are collected and made available in many places, most developing countries still struggle with limited availability due to inconsistent methodologies

and datasets (Pozzi et al., 2013; Smith et al., 2015; Trigg et al., 2016). This may for example be because of the lack of rain and discharge gauges due to insufficient resources as a consequence of socio-economic issues (Hughes, 2006; Spaliviero et al., 2011). One of the regions where data availability is poor is (Southern) Africa (Kundzewicz et al., 2002; Naumann et al., 2014; Trigg et al., 2016; UNISDR, 2016). Not only is there a general lack of data, but the available data and resources are also spatially asymmetrical (Thiemig et al., 2011). While the country of South Africa is relatively rich in terms of data, technology

and knowledge, many of its neighbouring countries are not (Spaliviero et al., 2011). This lack of spatially consistent datasets is a particular issue in this region, as many of the larger river basins are transboundary, and extreme events are often linked to phenomena on a wider, regional scale, such as cyclones (Biswas, 1999; Patt and Schro, 2008).

To address the issue of floods in data poor regions, increasingly available global datasets, such as global re-analysis data, may

have significant potential. Re-analysis datasets are the result of a combination of earth observations, as well as various models and datasets containing in-situ measurements (Schellekens et al., 2017). Currently there are several re-analysis datasets available at a global scale and applicable to water resources, such as ERA-Interim/Land (Balsamo et al., 2015), GLDAS (Rodell et al., 2004), Global Water Cycle Reanalysis (van Dijk et al., 2014), GSWP-2 (Dirmeyer et al., 2006), WATCH (Haddeland et al., 2011) and WRR (Schellekens et al., 2017). These datasets provide consistent hydro-meteorological data

with a global coverage, spanning several decades. Hence, they have significant potential to fill data gaps in regions such as Southern Africa (Sood and Smakhtin, 2015; Trigg et al., 2016; Ward et al., 2013; Wood et al., 2011). Datasets containing different global model outputs have thus far been used to determine climatic extremes at the global or continental scale. For instance, Zhao et al. (2017) evaluated the influence of different river routing schemes in the various global hydrological models



on peak discharge simulation. Dankers et al. (2014) compared the 30-year return period level of river discharge calculated using nine different global models regarding their projections of climate change impacts on flood hazards worldwide. Trigg et al. (2016) assessed the ability of six global models regarding their skill to produce hazard maps for the African continent. However, they note that there has thus far been limited validation of these global flood models against observed floods.

This study assesses the potential of a recently developed state-of-the-art global water resource re-analysis dataset in identifying damaging flood events for data poor regions such as the Limpopo River basin. The Limpopo River basin is a transboundary Southern African basin typical of the aforementioned data issues, including a general lack of data as well as an asymmetrical distribution of data availability across the riparian countries. The dataset used in this study is the open source global Water Resources Re-analysis (WRR) dataset developed in the eartH2Observe (E2O) research project, a collaborative project funded under the European Union's 7th Research Framework (Schellekens et al., 2017). The WRR dataset is assessed against two benchmarks. The first benchmark is developed using observed discharges from reliable gauges available in the region. As the upstream catchment area of these gauges varies, this provides insight into the skill of the global dataset in identifying the occurrence and magnitude of flood events in the basin, and how this skill is related to catchment scale. The second benchmark considers reported damaging flood events in the basin. Reported events from three disaster databases, including the Emergency Events Database (EM-DAT); the Global Active Archive of Large Flood Events (GAALFE); and the Natural Catastrophe Service (NatCatSERVICE), were collated to develop a chronology of damaging events. The ability of the global model datasets in identifying such damaging events provides insight into the potential of the global models to be able to predict the occurrence of impacting flood events. Global models such as those considered in this study are employed by several global (Alfieri et al., 2013) and continental (Thiemig et al., 2011) flood forecasting systems, and this assessments sheds light on the potential these have in providing information that is useful to managing floods at the local scale.

The remainder of this paper is structured as follows. Section 2 provides the materials and methods used, a description of the study area, data as well as verification methods. The results (Section 3) reveal the skill of the models in capturing the reported as well as modelled flood events. Section 4 provides a discussion of those results, as well as limitations and suggestions for further research. Conclusions are provided in Section 5.

## 2 Materials and methods

### 2.1 Study area

The Limpopo River basin is a transboundary river basin located in the east of Southern Africa, between latitudes 20°S – 26°S and longitudes 25°E – 35°E. With a length of approximately 4,000 km and a total drainage area of nearly 413,000 km$^2$, it is one of the larger basins in Southern Africa (Aich et al., 2014b; Maposa et al., 2014; Trambauer et al., 2015). The basin is shared by four riparian countries: South Africa, Botswana, Mozambique and Zimbabwe, as shown on Figure 1. The climate in





the basin is predominantly dry, semi-arid and hot (FAO, 2004; Trambauer et al., 2015). The upstream part is located in the Kalahari Desert, while further downstream, the climate transitions from an arid desert to a hot and dry steppe and eventually to a dry tropical savannah.

Precipitation in the basin varies significantly and is highly seasonal (FAO, 2004). Mean annual rainfall is approximately 530 mm, ranging between circa 270 and 1,160 mm (Beck et al., 2017). Some 95 % of the rainfall falls during the austral summer months between October and March, with the monsoonal rainfall events interspersed with dry spells. Precipitation events during the wet season are spatially as well as temporary isolated (FAO, 2004). The runoff ratio of the Limpopo River basin is low (Trambauer et al., 2014), which is characteristic for arid and semi-arid regions (Aich et al., 2014b), and is exacerbated in

the Limpopo basin by water abstractions for irrigation and domestic use. The basin faces significant transmission losses, resulting in a decline of flow along the length of the river (WMO, 2012). Large sections of the main stem, especially near the mouth, have a dry river bed during the dry season (LBPTC, 2010). However, flood waters can rise quickly, especially in the floodplains around Chokwé in Mozambique, where the mean flood peak can raise water levels some five metres above normal levels, with levels twelve metres above normal observed during the severe floods of the year 2000 (WMO, 2012). Furthermore,

the river basin has been modified to a large extent, with many dams, irrigation schemes, and storage reservoirs (Aich et al., 2014b; Ashton et al., 2001; LBPTC, 2010; Silva et al., 2010).

## 2.2 Input data

Input data in this research were provided by the publically available WRR dataset that was developed within the E2O research initiative. This dataset includes the outputs of ten different global models that are available at two resolutions and time ranges;

denoted WRR1 and WRR2. WRR1 has a 0.5 degree resolution (approximately 50 km at the equator) from 1979 to 2012 with the models forced by the Watch Forcing Data applied to ERA-Interim data (WFDEI) meteorological re-analysis dataset (Weedon et al., 2014). WRR2, on the other hand, has a 0.25 degree resolution from 1980 to 2014, and all models were forced using the Multi-Scale Weighted-Ensemble Precipitation (MSWEP) dataset (Beck et al., 2017). More information on the WRR dataset can be found in Dutra et al. (2015) and Schellekens et al. (2017).

As this research focusses on the occurrence of floods, simulated discharges of the ensemble of models included in the WRR datasets were used. Of the ten models, seven models provide daily discharge values, both Global Hydrological Models (GHMs) and Land Surface Models (LSMs). All apply different routing schemes to compute the discharges, see Table 1 for further information. The remaining three models do not include routing schemes and were therefore not considered. Discharge data

for both WRR1 and WRR2 were downloaded at the locations of the river gauging stations in the model-grid. While modelled discharges were available for evenly spaced grid cells, river gauging stations are not equally distributed across the Limpopo River basin, resulting in multiple gauging stations in the same model cell in some cases. The daily modelled river discharges for each cell in the model-grid where one or multiple discharge gauging stations is located were downloaded from the E2O





Water Cycle Integrator portal (https://wci.earth2observe.eu/). Modelled discharge data from the period 1980-2012 were used in this study as a common period in order to compare the differences between WRR1 and WRR2. Note that for four models simulated discharges were not available for the higher 0.25 degrees resolution models (see Table 1).

### 2.3 Verification data

### 2.3.1 Discharge data

Daily observed discharges from selected river gauging stations in the Limpopo River basin were used to verify the modelled discharges. Discharge records were collected from multiple sources and collated, including the Global Runoff Data Centre (GRDC), the South African Department of Water and Sanitation (DWAF) and the Regional Water Administration of Southern Mozambique (ARA Sul). In the entire Limpopo River basin, there are 196 accessible stations that contain data in the 1980 to 2012 time span. However, only 75 of these have daily data available for at least 25 years and passed the goodness of fit test by calculating the Kolmogorov Smirnoff statistic for the Gumbel Extreme Value Distribution at the 5% significance level. These 75 stations are shown in Figure 1, and a detailed list is included in the Supplementary material (S1). The stations have upstream catchment areas that vary between 4 and 342,000 km$^2$.

### 2.3.2 Disaster data

Data from three disaster databases were compiled in order to determine a singular chronology of damaging flood events in the Limpopo River basin to be used as a benchmark: EM-DAT (CRED & Guha-Sapir, 2017), GAALFE (Brakenridge, 2017) and NatCatSERVICE (Munich Re, 2017). This combined reference database contains the 48 damaging flood events that occurred in the basin over the time span that coincides with the period of record of the E2O dataset; from 1980 to 2012. A summary of this benchmark dataset is included in the Supplementary material (S2). To allow comparison of the reported events to the simulated and observed discharges, the severity or intensity levels of the reported damaging flood events were assessed. This was completed following the criteria from NatCatSERVICE (Kron et al., 2012), amended for the number of fatalities in the whole basin. This resulted in severity levels ranging from 0 (natural events) to 5 (devastating catastrophes).

The basin is both affected by large-scale basin-wide flood events, as well as by smaller scale flood events that do not affect the whole basin at once. The three disaster databases are structured differently. Whereas EM-DAT and NatCatSERVICE report the flood events on a country basis, the GAALFE is ordered on an event basis. Apart from that, the level of detail regarding the location of where the flood took place varies, also within one database. Especially the flood events that occurred earlier often have only a broad administrative descriptions, rather than the (sub-)basin of where the flood occurred. The study area was therefore subdivided into seven administrative regions in order to be able to make a spatial distribution in areas exposed to flooding. These regions are the Limpopo basin with the riparian countries Botswana (BW), Mozambique (MZ) and Zimbabwe (ZW), and four regions within South Africa (ZA). South Africa was split into multiple regions since roughly half



of the total basin area is located within South Africa, while nearly all of the available stations are within this part of the basin, allowing a higher level of detail in identifying the spatial occurrence of flood events. The four different regions in South Africa identified are the North West Province (ZA1), the Gauteng Province (ZA2), and the combined provinces of Limpopo and Mpumalanga, subsequently divided into a western (ZA3) and an eastern part (ZA4). The different regions can be seen in Figure

5   1.

## 2.4 Evaluating the model performance

### 2.4.1 Hydrological performance

Hydrological performance of the daily simulated discharges from all models was assessed using commonly used model evaluation statistics, considering Nash-Sutcliffe Efficiency (NSE), Percent Bias (PBIAS) and Pearson's correlation coefficient

(r). For a fuller description of these statistics and their application see Moriasi et al. (2007). NSE ranges between $-\infty$ to 1, where 1 indicates a perfect representation of observed discharges, with values above zero meaning the simulated discharges have better skill than simply taking the average of the observed. PBIAS determines the tendency of the simulated discharge to underestimate or overestimate observed discharges (Gupta et al., 1999), normalised with the mean discharge. Ideal values of PBIAS are zero, with acceptable values considered to be below ±25 percent (Moriasi et al., 2007). Pearson's correlation

coefficient (r) provides an indication of the linear relationship between simulated and observed discharges data. Ranging from −1 to 1, which indicate a perfect negative or perfect positive relationship respectively, a correlation coefficient of 0 shows no relationship whatsoever. Correlation coefficients are widely used to describe the proportional decrease or increase of two variables, and have the advantage to be sensitive to large values (Beck et al., 2017; Legates & McCabe Jr., 1999), which is important for analysing hydrological extremes.

### 2.4.2 Hydrological extremes

#### Flood Frequency Analysis

Flood frequency analysis was performed in order to obtain the magnitudes of the hydrological extremes (Mujere, 2011). By fitting a Gumbel distribution using the method of moments, the daily river discharge values were converted to annual exceedance probabilities or return periods (Ward et al., 2011). This allows the occurrence and severity of flood events to be

identified in both the observed and modelled discharge time series. Observed flood events were identified as events with a low annual exceedance probability (or high return period) at the river gauging stations, with discharges associated to progressively smaller probability thresholds used to identify increasingly severe flood events. Flood events in the modelled discharge time series were identified in two ways; using either the model climatology or the observed climatology. When using the model climatology, the discharge values for the selected probability thresholds were derived using the Gumbel distribution applied

to the modelled discharges, providing the skill of the model in simulating the variability of extreme discharges. When using the observed climatology, the discharge values for the thresholds were derived using the observed discharges, which represents



the skill of the model in determining the absolute discharges. The severity of the reported damaging flood events retrieved from the three disaster databases (Section 2.3.2) are then compared to the severity of the flood events identified in observed and modelled time series. To allow this comparison, the reported damaging flood events, the annual exceedance probabilities or return periods were converted to flood intensity levels, according to Table 2. In order to determine the possible added value of the higher resolution global models, modelled flood events were assessed both for WRR1 and WRR2, as well as for each of the individual models, and the model ensemble.

**Skill Scores**

The ability of the models to detect the flood severity was assessed using a contingency table in combination with three skill scores that were based on the model climatology and derived from the table as performance measures. The annual exceedance probabilities (or return periods) for the measured discharge and modelled discharge extracted from the model-grid cell corresponding to the location of the gauge, were computed using the Gumbel distribution which was estimated using the method of moments. A moving window of seven days for both the observed as well as the modelled discharge was applied to select the maximum discharge of a given event. This window was chosen to disregard possible small time lags between the modelled and observed discharges (Thiemig et al., 2012). The annual exceedance probability thresholds were then used to assess whether or not the modelled discharge is able to capture the timing and intensity of the extreme events. To compare the relative performance of the models, different annual exceedance probability thresholds were used for the modelled as well as for the observed discharges, ranging between 0.342 and 0.005, equivalent to return periods of 1.5-year and 200 years, respectively. These thresholds were used to establish the contingency table for the observed discharge at each gauging station with the discharge from its matching model cell, as shown in Table 3. The table identifies the hits (H, flood events are both modelled and observed in the gauged data), misses (M, flood events are observed but not modelled), false alarms (FA, flood events are modelled but not observed) and correct negatives (CN, flood events are neither observed nor modelled).

Skill scores to quantify the ability of the models to identify flood events were derived from these contingency tables, and include the Critical Success Index (CSI), the Probability of Detection (POD) and the False Alarm Ratio (FAR). These were assessed for each model using either the model or the observed climatology. The CSI and POD determine the percentage of successfully forecasted events of all events observed, whereas the FAR identifies the percentage of incorrectly forecasted flood events out of all events forecasted. The ideal value for CSI and POD is at 100%, while for FAR it is at 0%. The CSI, POD and FAR are calculated using Equations 1, 2 and 3:

$$CSI = \frac{H}{H+M+FA} * 100 \tag{1}$$

$$POD = \frac{H}{H+M} * 100 \tag{2}$$

$$FAR = \frac{FA}{H+FA} * 100 \tag{3}$$





### 2.4.3 Damaging hydrological extremes

The capability of the models in capturing the flood events that resulted in reported damages was illustrated graphically. The relationship of the severity levels of the damaging flood events that were reported by the disaster databases, and the corresponding annual exceedance probabilities of the observed as well as the modelled discharges at the gauging stations was

illustrated. For each reported event, the corresponding maximum discharge (and thus the lowest annual exceedance probability) in either the observed or simulated time series was determined with a moving average of three days before and after the start and end date of the reported flood event (corresponding to a window of seven days for flood events reported to occur on a single date). The reported damaging flood events are reported as occurring in one or more of the defined regions. However, as the disaster databases typically report only the broad administrative region of where the flood took place, there was often not

enough information available on the sub-basin scale. Therefore, to associate the reported flood events in a region to a flood event being identified in either the observed or the modelled discharges, the lowest annual exceedance probability for every event was determined for each observed river gauging station and corresponding model-grid cell in WRR1 for all stations with an area larger than 2,500 km$^2$, and in WRR2 for all stations with an area larger than 520 km$^2$. These sizes of the catchment areas for WRR1 as well as WRR2 were assessed using the NSE statistic in Section 3.1. This process was repeated for all events

and for every region in the basin.

### 3 Results

### 3.1 Hydrological performance

The relationship between the upstream catchment area of the river gauging stations in the Limpopo River basin and the error statistics for the models in WRR1 and WRR2, is illustrated in Figure 2 and Table 4. Figure 2 and Table 4a show the three

models that are available both in WRR1 and WRR2, whereas Table 4b also provides the performance statistics for the models that are available only in WRR1, as well as the results using the mean of the seven-member ensemble based on the models in WRR1. The different results demonstrate the improvement of model simulations for stations with a large upstream catchment area, when compared to those with smaller ones. This can be best observed by looking at the NSE statistic, from which it is evident that the models are generally able to capture the hydrology for stations with an upstream catchment area that is larger

than 2,500 km$^2$ for WRR1 (Figure 2a), and larger than 520 km$^2$ for WRR2 (Figure 2d). This provides an indication of the catchment size at which the models are capable of capturing the hydrology, and also illustrates the difference of the higher resolution WRR2 as compared to WRR1. The NSE values show that for WRR1 as well as for WRR2, the HTESSEL-CaMa and WaterGAP3 models both perform reasonably well and had roughly equal NSE values, even though the structure of the models is quite different, as the former is a land surface model, while the latter is a global hydrological model.






PBIAS (Figure 2b and 2e, and Table 4) largely shows negative values, indicating an overestimation of the models compared to the observed discharges. This overestimation is visible for all models, and is more dominant at the stations with the smallest upstream catchment areas. This can be expected, as the models take the discharge accumulated over a large area (approximately 2,600 km$^2$ and 650 km$^2$ for WRR1 and WRR2 respectively) as the value for one model-grid cell, whereas the true upstream catchment areas of the stations may be as small as 4 km$^2$. The models for which the overestimation is lower, and which thus generally perform better, are again HTESSEL-CaMa and WaterGAP3, both in WRR1 and WRR2. Furthermore, HTESSEL-CaMa is the only model that frequently under predicted the discharges, reflected by a positive PBIAS value. The seven models and model ensemble mean that were available in WRR1 (Table 4) have quite distinct differences. The models ranked from best to worst for NSE and PBIAS were; HTESSEL-CaMa; WaterGAP3; SURFEX-TRIP; the ensemble mean; ORCHIDEE; LISFLOOD; PCR-GLOBWB; and W3RA. The poor performance of W3RA was attributed by consistent severe overestimation of modelled discharges.

The last error statistic considered is Pearson's correlation coefficient, r, displayed in Figure 2c, Figure 2f and Table 4. This error statistic shows relatively consistent correlations for each model, irrespective of the upstream catchment areas. For WRR1, the models that performed best are respectively; LISFLOOD; SURFEX-TRIP; the model ensemble mean; and WaterGAP3, whereas the poorest performance is found for PCR-GLOBWB, and to a lesser extent HTESSEL-CaMa. For WRR2, WaterGAP3 performs significantly better, and also the improvement of WRR2 over WRR1 is notable for both WaterGAP3 and HTESSEL-CaMa. LISFLOOD, on the other hand, has a lower r value for WRR2 compared to WRR2. The WRR1 models scores differently for the r values when compared to ranking for NSE and PBIAS. The order, ranking from best to poorest order is; LISFLOOD; SURFEX-TRIP; the ensemble mean; WaterGAP3; W3RA; ORCHIDEE; HTESSEL-CaMa; and lastly PCR-GLOBWB.

Even though some models perform relatively well, the overall performance of the models is, however, quite poor. Average NSE remains negative for all models and upstream catchment areas. Average PBIAS was below 25% in only a few instances for the models HTESSEL-CaMa and WaterGAP3, and the average r value rarely exceeded 0.5.

## 3.2 Hydrological extremes

### 3.2.1 Flood Frequency Analysis

The ability of the models in predicting hydrological extremes was analysed by comparing the modelled hydrological extremes to the hydrological extremes that were observed at the river gauging stations (Spookspruit and Limpopo River), as well to the chronology of reported damaging flood events. Results are illustrated for two stations selected as an example in Figure 3. Modelled extremes were analysed using the discharge thresholds derived from either observed climatology or the modelled climatology. The locations of the two river gauging stations are shown in Figure 1, and the model selected is the WaterGAP3





model. Similar patterns were observed at stations with similar sizes and for the other models. Comparing the pattern of flood events identified by MM1 (WRR1 using the modelled climatology), as well as by MM2 (WRR2 using the modelled climatology) to the observed (Obs) or reported (Rep) flood events, it is clear that the WaterGAP3 model is relatively well capable of capturing the variation of the discharge in the observed data, as well as the occurrence of reported damaging events,

particularly at the station with a large upstream catchment area, though even at the station with a small upstream catchment area the correspondence in the patterns is reasonable.

Another result derived from Figure 3 is the ability of the models to capture the actual intensity of the identified flood events. This is indicated in the bottom two lines; MO1 and MO2, in which the severity thresholds were established using the model

climatology. The frequency of flood events for WaterGAP3 is quite a bit higher than the observed frequency, with the severity when observed and simulated events do line up also being quite a bit higher. This is clearly the result of the over-prediction of observed discharges. However, there is a marked improvement from the station situated in the river with a small upstream catchment area to the station with a large upstream catchment area, as well as when comparing the higher resolution WRR2 to WRR1. Similar results were found for other models and stations pairs, and accordingly also in the model performance statistics

discussed in the previous section.

### 3.2.2 Skill scores

The upper panel in Figure 4 shows the CSI for each of the models in WRR1, as well as for the seven-model ensemble mean, with discharge thresholds based on model climatology. The score for the models in WRR1 was found to be quite constant for discharges that occur more frequently, i.e. Annual Exceedance Probabilities higher than 0.09, equivalent to a return period of

10 years. The relative performance of the models from best to worst for these discharges is W3RA; the ensemble mean; SURFEX-TRIP; LISFLOOD; WaterGAP3; PCR-GLOBWB; HTESSEL-CaMa; and ORCHIDEE. The pattern, however, changes for the more extreme (low probability) discharges. The discharges with an annual exceedance probability that was less than 0.09 showed a greater spread, as well as changes in the order of performance of the models. For example, SURFEX-TRIP and LISFLOOD now perform better, while W3RA performs worse for these more extreme discharge events. The model

ensemble mean though has a remarkably high CSI score which is independent of the return period.

The differences in performance as a result of the increased spatial resolution in WRR2 becomes evident from the lower panel of Figure 4. For WaterGAP3 and HTESSEL-CaMa, the improved resolution results in higher CSI values. For LISFLOOD, on the other hand, the performance of WRR1 is better than that in WRR2. Again, it appears that WaterGAP3 WRR2 performs

best overall. These same patterns are observed regarding the error statistics, as shown in Figure 2 and discussed in Section 3.1.

The underlying reason for the observed patterns of the CSI can be explained by taking a closer look at the POD and FAR. The performances of all three skill scores with respect to the upstream catchment area of each individual station are shown in Figure



5. Skill scores are shown here for events with an annual exceedance probability of 0.164, equivalent to a return period of 5 years. As can be observed by looking at the models in WRR1 (upper panel), the average POD is around 25% and the average FAR is around 70%, resulting in an average CSI of roughly 15%. The CSI, POD and FAR all have a relatively large spread, with little relationship to the upstream catchment area of the stations. Stations with a larger upstream catchment area do not necessarily result in better skill scores. An explanation for the lack of a relationship with catchment areas is that the three skill scores are based on model climatology, and thus the relative flood intensity, while the error statistics are based on the observed climatology and thus the absolute intensities. This clarifies the notable difference with the error statistics, such as the NSE (as shown in Figure 2a and d), where the improvement of stations with a larger upstream catchment area is clear. This suggests that the performance of the models in estimating the relative intensity, is not highly influenced by the upstream catchment area of the river gauging stations. The difference in performance between WRR1 and WRR2 is, however, apparent. Both HTESSEL-CaMa and WaterGAP3 display improved values for the CSI, POD and FAR. Again, the notable exception is LISFLOOD, where WRR1 performs better than for WRR2, independent of the skill score. This again reflects the error statistics discussed in Section 3.1.

### 3.3 Damaging hydrological extremes

Scatter plots were used to demonstrate the relationship between the reported severity of the reported flood events with the severity of the corresponding events identified in the observed as well as the modelled discharges. These scatter plots are shown in Figure 6, illustrating the reported flood severity in discrete classes (x-axis), as well as the annual exceedance probability for the events identified using the maximum of the modelled or observed discharges in a seven day window around the reported event (y-axis). The exceedance probability found at each station is plotted. Ideally the events should be clustered along the diagonal from top left (higher probability, lower severity), to bottom right (lower probability, higher severity), reflecting that lower impact flood events typically occur in only a few stations and have higher probabilities (low return periods), while high impact severe flood events are often basin wide, occurring at most stations across the basin with lower probabilities. For medium severity reported events, a wider scatter would be expected, as these events may occur only in a part of the basin.

The figure shows that when a reported flood event is classified at the most severe category 5, impacts were observed throughout the basin, as all observed as well as modelled probabilities indicate above normal river discharge, many of which with extreme (low probability) discharges. Small-scale flood events that resulted in low as well as localised damages, on the other hand, were classified either as category 0 or 1. As can be seen in Figure 6, the annual exceedance probabilities corresponding to these events have a larger spread. The reason for this is that small-scale events are not noticeable throughout the entire region, but only locally, as many gauges were still measuring normal flow, while those where the event does occur show more extreme discharges. It can be observed though that part of the gauges measured an above normal discharge, whereas this was frequently not observed by the models. Only WaterGAP3 was able to detect extreme discharges for the floods with a severity level of



zero. Apart from that, the three different models displayed comparable results, although HTESSEL-CaMa generally had lower annual exceedance probabilities for the same flood events when compared to LISFLOOD and WaterGAP3.

## 4 Discussion

The potential of the global Water Resources Re-analysis dataset was assessed by studying the hydrological performance,
identification of hydrological extremes, as well as of damaging flood events, and were evaluated by means of commonly used error statistics and verification skill scores. The verification of the models within the WRR dataset was largely dependent on the observed river discharge data. Access to these data proved to be quite challenging, and the quality of the discharge data that was obtained was often insufficient. Only 75 of the 196 river gauging stations for which at least some data available in the Limpopo for the desired time range were used in this research, with most of these in South Africa. This has implications for
the conclusions drawn from the research, especially for the PBIAS as it is highly influenced by the uncertainty in the observed data (Moriasi et al., 2007). Despite these limitations, this research shows that the discharges that were estimated by the different global models are to some extent able to capture the variability of observed discharges, as indicated by the different error statistics. For instance, the NSE demonstrated that for WRR1 as well as WRR2, both the HTESSEL-CaMa and the WaterGAP3 models performed well with roughly similar NSE values, despite the different structure of these models. HTESSEL-CaMa is
a LSM and does not include lakes and reservoirs or water usage, whereas WaterGAP3 is a GHM and does include both lakes and reservoirs, as well as water usage (Table 1). The differences between the model structures is illustrated by the PBIAS and r values. HTESSEL-CaMa has reasonable PBIAS, while WaterGAP3 has a relatively good r. As noted in Section 2.1, the basin is highly altered due to human influences, in particular by a large number of storage reservoirs. Models that capture only natural flow conditions, and do not take the reservoirs and water usage into account, may be able to reasonably estimate runoff
volumes, though they do tend to largely overestimate the actual magnitude of the discharges. Not including human influences such as regulation, however, results in low correlations. The relative intensity of flood events, on the other hand, can still be well captured by the same model when using the model climatology instead of the observed climatology as a reference. An example of such a model is W3RA, which performs poorly when considering the error statistics, but relatively well for the CSI.

Global models are best suited for the modelling of large-scale processes, but poorly represent the small-scale ones such as the variability associated with convection (Beck et al., 2017). These conclusions have been drawn in similar research, such as Asante et al. (2008), Thiemig et al. (2015) and Trigg et al. (2016). This study indicates that the small-scale flood events were generally not well captured by the global models that were analysed in this research. The results do show, however, that the
performance of these global models can improve as the resolution of the global models improves. The statistics for model performance measures for the higher resolution WRR2 starts to approach reasonable values for gauges with upstream areas of some 500 km$^2$, while for the lower resolution WRR1 these same values are attained only at areas of some 2500 km$^2$. The higher



resolution WRR2 also shows for two of the three models better skill in identifying reported flood events, represented in the chronology of reported flood events developed. Whether the improved performance of the higher resolution is due to the improved and higher resolution MSWEP forcing data (Beck et al., 2017), or due to improved representation of hydrological processes is unclear. However, as the improvements vary between the models, it is clear that model structure has an influence.

That there is skill in these global models in identifying flood events that have impacts, and that this skill improves as the resolution of these large-scale models improves, is significant. Global scale forecasting systems (Alfieri et al., 2013) as well as those at continental scale (Thiemig et al., 2011) typically employ such large-scale models for developing forecasts, using thresholds based on model climatology to inform the severity of predicted events and subsequent issuing of flood warnings.

Such warnings may be issued where there are no (reliable) river gauges, as is the case in much of the Limpopo basin, making calibration of a local model difficult. The ability of these global and or continental models to predict the occurrence of flood events that have impacts bolsters the confidence of using these warnings to initiate response, though the high false alarm rate found could again diminish confidence.

It is important to note, though, that likely not all small-scale flood events that occurred between 1980 and 2012 will have been included in the chronology of reported flood events that was developed. As has more often found to be the case in the Global South (Dartmouth Flood Observatory, 2017), the availability of disaster data in the Limpopo River basin is fairly limited. In order to construct a basin-wide timeline of historic damaging floods, events reported in the EM-DAT, GAALFE and NatCatSERVICE databases were collated. Even though the three used here are currently the most comprehensive databases containing reported damaging historic flood events in Africa (Aich et al., 2014a), several shortcomings are noted. These include

inconsistencies between events reported, gaps, and limited reporting in some areas (Guha-Sapir et al., 2016). Additionally, most disaster databases are available at the country scale, whereas flood events occur at the basin scale. It is recommended to enhance the reporting of flood disasters by providing more details on the losses that were incurred as well as a more precise description of the location and extent of the floods. The basin-wide approach to identify past flood events by using empirical

disaster databases used in this research has also been applied in other research (Aich et al. (2014a), Asante et al. (2008), Bischiniotis et al. (2017), Huggel et al. (2015) and Thiemig et al. (2015)), noting similar deficiencies.

The flood classification that was used in this research is a discrete classification, taking the number of fatalities and overall losses into account. However, it is expected that a continuous flood severity classification would be better able to reveal the

relationship between extreme river discharges and the intensity of reported damaging flood events. However, due to the gaps in the reported damaging flood event data as well as broad area descriptions, this could not be assessed at this point. In order to identify the added value of such a classification, additional research is required in addition to improving disaster loss data.





In this study, the Gumbel distribution is used to determine the annual exceedance probability thresholds of both the modelled and observed discharge data. Different extreme value distribution, however, can significantly influence the probability of the extreme discharges (Dankers and Feyen, 2008). The Gumbel distribution is a two-parameter distribution and was applied due its simplicity and robustness, though some authors (e.g. Ponce (1989), argue that a three-parameter distribution such as the GEV or the Log Pearson type III should be used for flood frequency analysis. However, the goodness of fit of the distributions found was tested using the Kolmogorov Smirnoff test, with stations that did not meet the 5% significance threshold not considered. Further inspection of these stations revealed that these were often directly downstream of a dam, or otherwise strongly influenced by human activities. Additional research could additionally explore the influence of using more complex extreme value distributions. This could also consider the influence of the length of the moving window that was used to identify the maximum discharge in the observed and modelled time series. This moving window was chosen to allow for the travel time from the upstream parts of the sub catchment. In reality, however, the catchment upstream of each gauging station has its own time of concentration, and the window used could be made specific for each station accordingly.

Of the global models considered in this study, the higher resolution WaterGAP3 in WRR2 demonstrated the best performance, both for capturing the hydrological behaviour across the Limpopo basin, as indicated by good values for the error statistics, as well as for identifying the occurrence and severity of hydrological extremes, which was indicated by the skill scores. It was also observed that WaterGAP3 in WRR2 is reasonably good at estimating low annual exceedance probabilities for the damaging flood events for the stations with a large upstream catchment area. One reason for this improved performance may be the inclusion of lakes and reservoirs, as well as water abstractions in the model. However, results for other models, such as W3RA, which has the worst model performance error statistics, may rank higher than other models when used to identifying the occurrence of flood events, where these are identified using the model's own climatology. It is also important to note that if similar research would be applied elsewhere, the ranking of model performance may be quite different. The ranking of the models also clearly depends on the aim of the research. For instance, when the key interest is the relative performance of the model for the Limpopo River basin, taking only the model climatology into account, the W3RA model would be the preferred model, as it has a high CSI. However, when the main goal would be the absolute magnitude of discharges, the W3RA model would not be considered, as it is found to severely overestimate the discharges in the Limpopo River basin. The seven-model ensemble mean, on the other hand, proved to be quite consistent in its performance. For the CSI values particularly it scores remarkably high, but it also scores relatively well for the Pearson's correlation coefficient r. Though it should carefully be assessed which model would be the best applicable for each instance, the model ensemble mean would be the safest bet in an area where no model clearly stands out.



## 5 Conclusion

The study explores the use of a global re-analysis dataset developed within the EU-FP7 EarH2Obverve (E2O) project, which is constructed using a set of global hydrological and land surface models, to support flood risk analysis in data sparse regions, such as the Limpopo River basin. There is a necessity for such re-analysis data, since measured river discharge data in this basin and others like it are currently insufficient, poorly spatially distributed, have an insufficient period of record, or are partly inaccessible. The E2O re-analysis dataset provides hydro-meteorological data of sufficient length and coverage required for statistical analysis. When the variability of the discharge results of the ensemble of models included in the re-analysis dataset is evaluated, the error statistics found show that the models all have reasonable skill in capturing the variability of the observed discharges, though there may be significant bias in magnitude. This was indicated by strong correlations, low Nash-Sutcliffe Efficiency and high percent-bias values. Furthermore, the error statistics revealed that the variability is better captured by the models at hydrological gauging stations that have larger upstream catchment areas compared to those in smaller catchments. The upstream catchment areas of the river gauging stations at which WRR1 and WRR2 are able to provide representation of the hydrological behaviour that is better than the average of the observed is found for catchment areas of some 2,500 km$^2$ and 520 km$^2$ and above respectively, with significantly poorer performance for smaller catchment sizes. This shows that the continued improvements in the global models with a higher resolution, either due to improved higher resolution forcing, or due to improved model structures, can be expected to lead in most cases to better capabilities of capturing the variability of the observed discharge as well as the magnitude of observed discharges.

A novel aspect of this study is in exploring the skill of the global models in identifying the occurrence and severity of flood events in two benchmark chronologies of flood events. The first was developed through flood frequency analysis, with flood events identified to occur at selected probabilities, while the second was developed through collating reported flood events in three disaster impact databases. This shows that the global models do have skill in capturing the observed as well as reported damaging events. This is, however, only the case when the thresholds of the discharges corresponding to the flood events are determined using the model climatology, and not the observed climatology. The simulated discharges of these global models are thus found to better represent the variability of the observed discharges, than the magnitude; though this is less an issue for the better performing higher resolution models of WRR2.

Despite the absence of high-quality data in the Limpopo River basin and the coarse resolution of the models in the global re-analysis dataset, this research shows that regardless these limitations, the global re-analysis dataset can provide valuable information for flood risk assessment in data sparse regions. The skill of the models to predict flood events in the basin that have led to flood damage, as recorded in the chronology of reported floods is an important finding, as global models such as those assessed here are often used in global and continental forecasting systems to generate flood forecasts and issue warnings in basins with little or no gauged data, but where floods and consequent impacts do occur. This indicates that openly available




global scale hydro-meteorological data can provide valuable information regarding extreme events in data sparse regions and may therefore be of use to local decision makers in mitigating the negative consequences of future flood events, and that this may improve as the resolution of these global models improves.

**Acknowledgements**

This work received funding from the European Union Seventh Framework Programme (FP7/2007-2013) under grant agreement no. 603608, Global Earth Observation for Integrated Water Resource Assessment (eartH2Observe). We are grateful to Munich Re for providing historic flood events from their NatCatSERVICE database in the framework of NWO VIDI grant 016-161-324, and EU FP7 IMPREX project, grant agreement no. 641811.

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

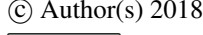



**Figure 1:** Map of Limpopo River basin with the riparian countries, major tributaries, and the seven regions in the basin that were identified for this research. Also shown are the river gauging stations with at least 25 years of data between 1980 and 2012. The square is located upstream in the Spookspruit tributary (252 km²), and the circle is at the main stem of the Limpopo River (98,240 km²).





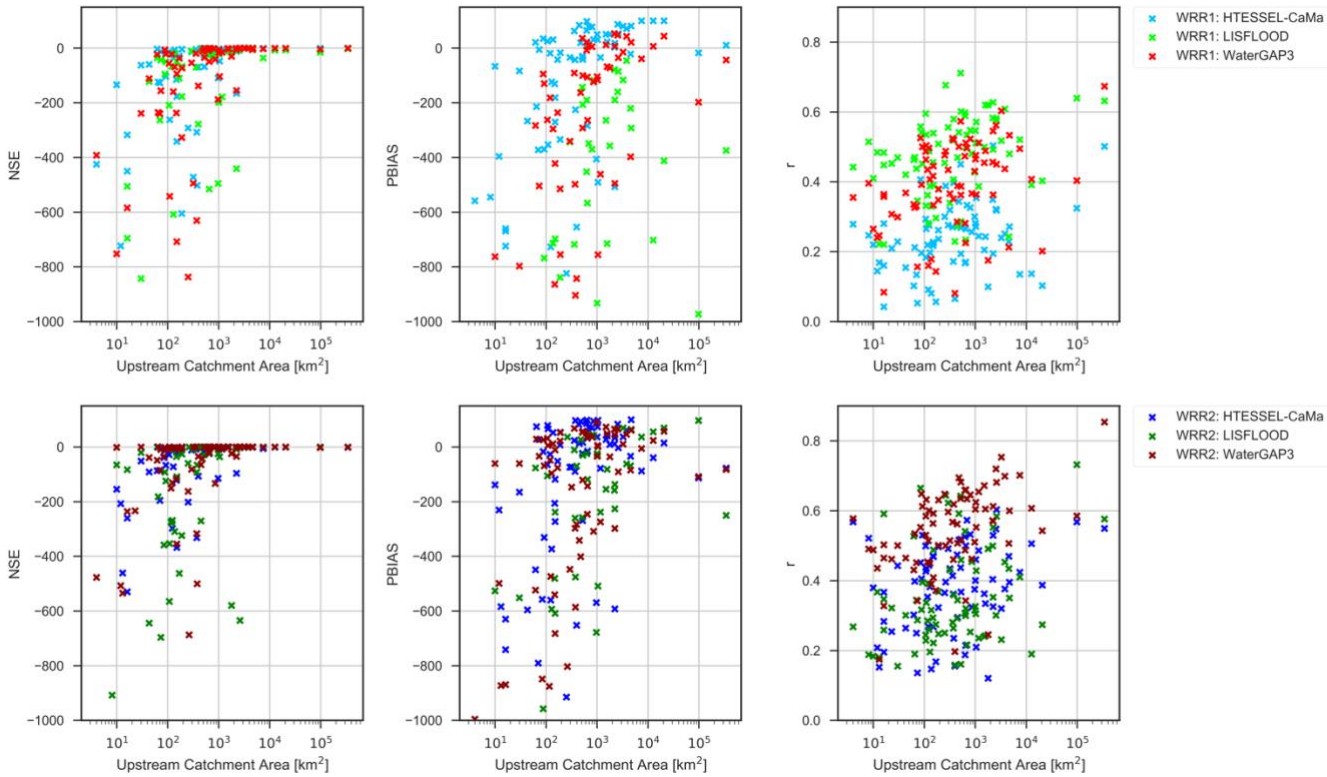

**Figure 2:** Performance statistics for the three models available in both WRR1 (top) and WRR2 (bottom) for each of the 75 gauging stations in the Limpopo River basin, ordered by upstream catchment area. The error statistics displayed include (a) the Nash Sutcliffe Efficiency (NSE) for WRR1, (b) the Percent Bias (PBIAS) for WRR1, (c) Pearson's r for WRR1, (d) NSE for WRR2, (e) the PBIAS for WRR2, and
5   (f) Pearson's r for WRR2. For clarity, the lower limit of the y-axis of the NSE and PBIAS has been set to -1,000.





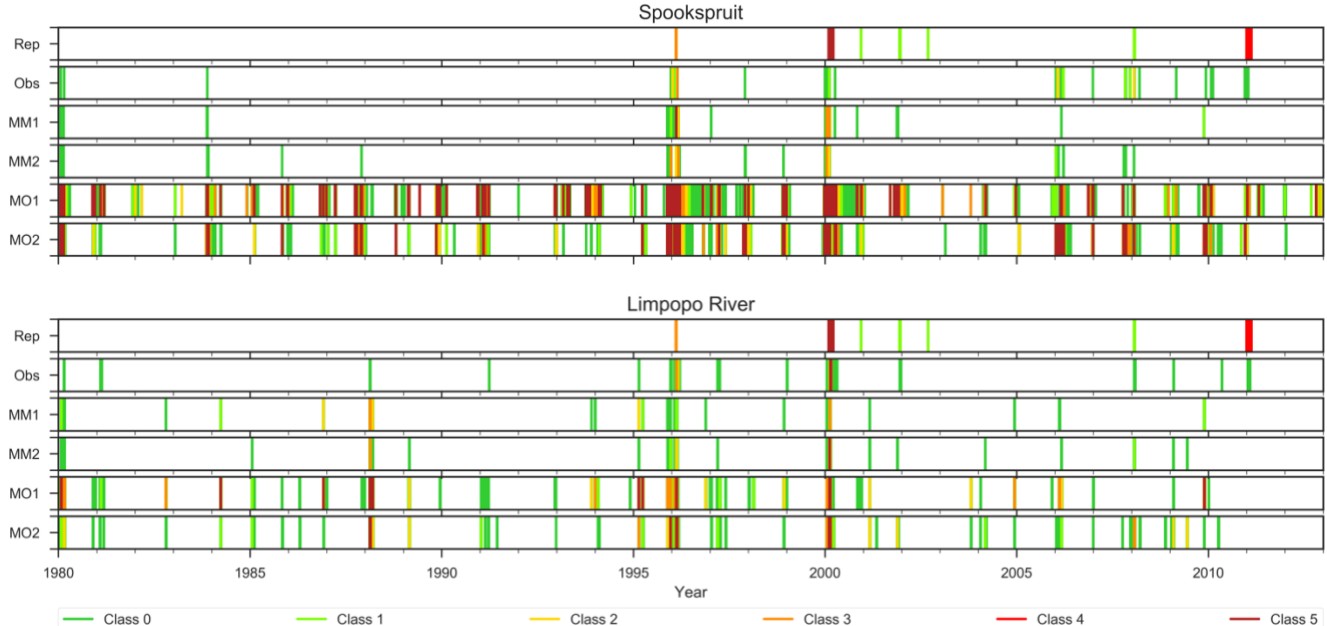

**Figure 3:** Occurrence of flood events of increasing severity classes at the Spookspruit gauge (252 km$^2$; upper panel) and in the main Limpopo River (98,240 km$^2$; lower panel). Model flood events were identified using model climatology (MM1 & MM2) or observed climatology (MO1 & MO2), and were compared to benchmarks based on a compiled disaster impact database (Rep) and observed river discharge data (Obs). The index value refer to models with 0.5 degree resolution (1) and 0.25 degree resolution (2). Results are shown for the WaterGAP3 model, which is available in the eartH2Observe Water Resources Re-analysis dataset.





**Figure 4:** The Critical Success Index (CSI) using different annual exceedance probability thresholds averaged over all gauging stations for the seven models and ensemble mean available in WRR1 (upper panel), and the three models that are also available in WRR2 (lower panel).





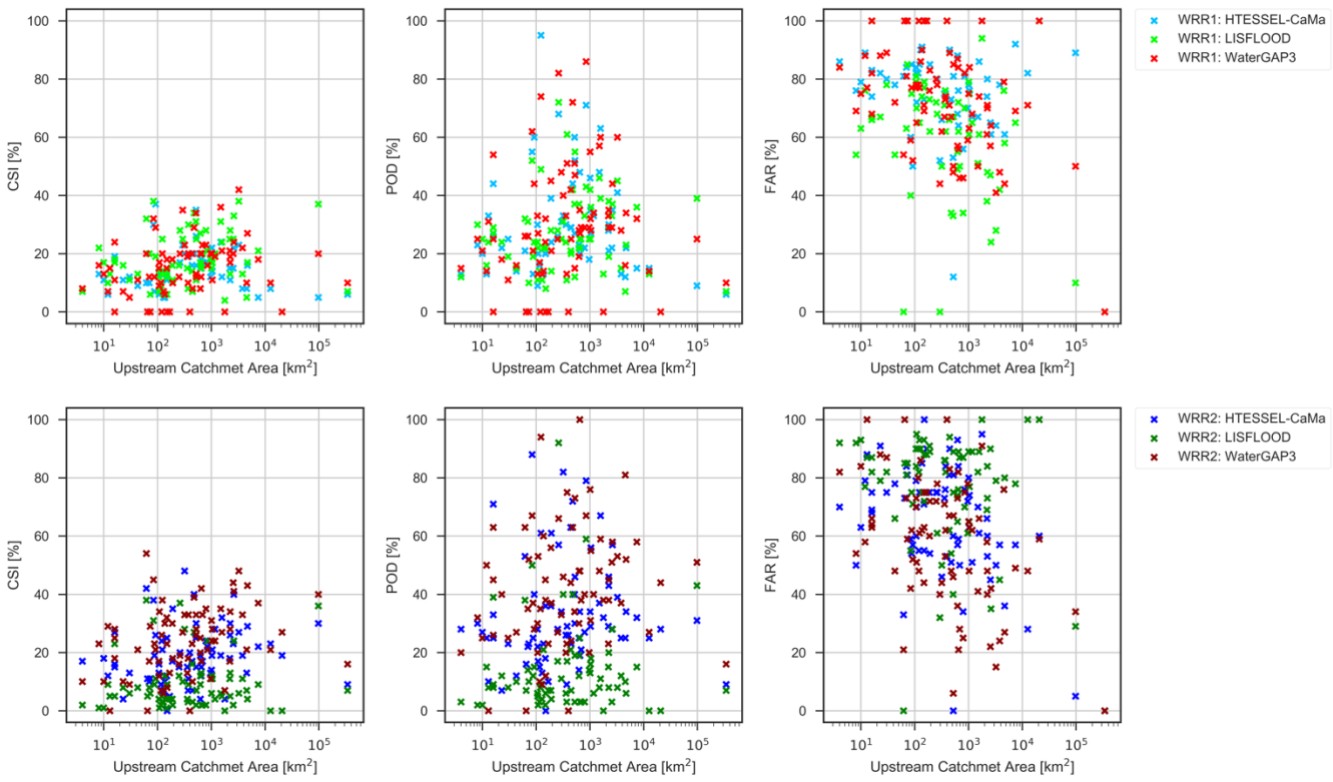

**Figure 5:** The Critical Success Index determined using the annual exceedance probability threshold of 0.164 (return period of 5 years) for all gauging stations for the three models available in WRR1 (upper panel), and the models that are also available in WRR2 (lower panel).





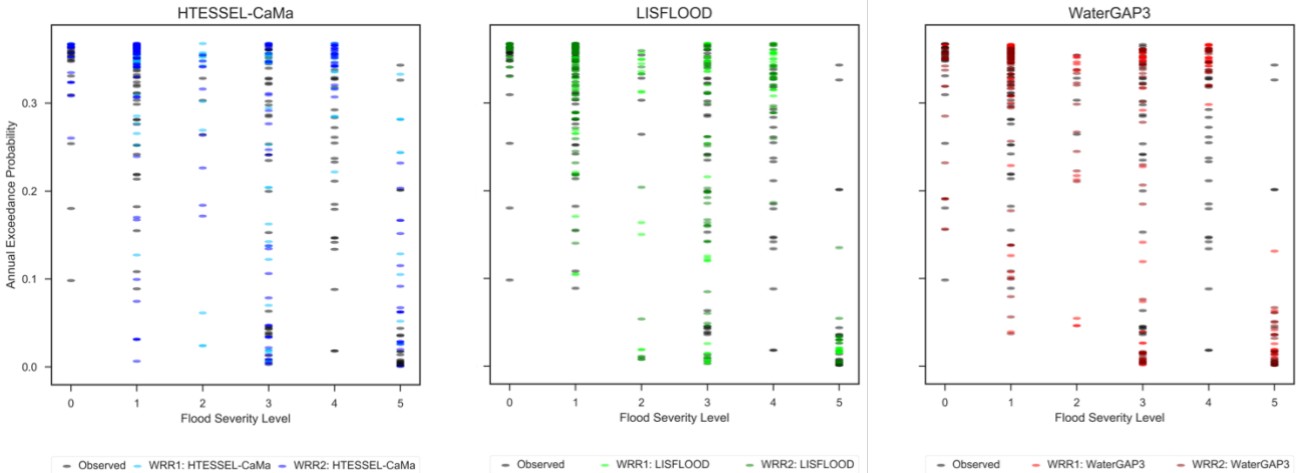

**Figure 6:** The relationship of the flood event severity for the reported flood events, and the corresponding annual exceedance probabilities that were observed and modelled for (a) HTESSEL-CaMa, (b) LISFLOOD, and (c) WaterGAP3.



**Table 1:** Overview of the seven global models in the Water Resources Re-analysis dataset that include daily river discharges

| Model | Model Type | Resolution (degrees) | Lakes-Reservoirs | Water use | Routing | Reference |
|---|---|---|---|---|---|---|
| HTESSEL-CaMa | LSM | 0.5 & 0.25 | No | No | CaMa-Flood | (Balsamo et al., 2009) |
| LISFLOOD | GHM | 0.5 & 0.25 | Yes | Yes | Double kinematic wave | (van der Knijff et al., 2008) |
| ORCHIDEE | LSM | 0.5 | No | No | Linear cascade of reservoirs | (Krinner et al., 2005) |
| PCR-GLOBWB | GHM | 0.5 | WRR1 only lakes | Not in WRR1 | Travel time | (van Beek and Bierkens, 2009) |
| SURFEX-TRIP | LSM | 0.5 | No | No | TRIP with stream | (Decharme et al., 2010) |
| WaterGAP3 | GHM | 0.5 & 0.25 | Yes | Yes | Manning-Strickler | (Flörke et al., 2013) |
| W3RA | GHM | 0.5 | No | No | Cascading linear reservoirs | (van Dijk et al., 2014) |
| Ensemble 7 models | GHM & LSM | 0.5 | Various | Various | Various | N/A |

[*Source*: Schellekens et al. 2017; Dutra et al., 2015]





**Table 2:** Thresholds that were used to classify the exceedance probabilities according to flood severity levels

| Flood Severity Level | Annual Exceedance Probability | Return Period [years] |
|:---:|:---:|:---:|
| 0 | ≤ 0.303 | ≥ 2 |
| 1 | ≤ 0.164 | ≥ 5 |
| 2 | ≤ 0.090 | ≥ 10 |
| 3 | ≤ 0.038 | ≥ 25 |
| 4 | ≤ 0.010 | ≥ 100 |
| 5 | ≤ 0.005 | ≥ 200 |

**Table 3:** Contingency table for flood events

| | | Observed | |
|:---:|:---:|:---:|:---:|
| | | **Yes** | **No** |
| **Modelled** | **Yes** | Hits (H) | False Alarms (FA) |
| | **No** | Misses (M) | Correct Negatives (CN) |

5  [*Source*: Thiemig et al., 2015]





**Table 4:** Performance statistics for the three models available in 0.5 degree WRR1 and 0.25 degree WRR2 (Figure 3, upper panel) for each of the 75 gauging stations in the Limpopo River basin, and the four models and ensemble mean only available in 0.5 degree WRR1 (Figure 3, lower panel). Statistics displayed for WRR1 and WRR2 include the Nash Sutcliffe Efficiency (NSE), the Percent Bias (PBIAS) and Pearson's r (r). The three different upstream catchment areas indicate that an average is taken of the error statistics of the stations that are larger than indicated. Stations ≤ 4 indicates all 75 stations, ≤ 520 are the largest 31 stations, and ≤ 2,500 are the largest 11 stations.

| Stations with upstream catchment area [km²] | | HTESSEL-CaMa | | LISFLOOD | | WaterGAP3 | |
|---|---|---|---|---|---|---|---|
| | | WRR1 | *WRR2* | WRR1 | *WRR2* | WRR1 | *WRR2* |
| **NSE** | ≤ 4 | -734.77 | *-294.83* | -6,473.49 | *-23,616.81* | -1,445.34 | *-628.78* |
| | ≤ 520 | -16.38 | *-9.62* | -107.99 | *-43.12* | -21.94 | *-8.16* |
| | ≤ 2,500 | -0.16 | *-0.82* | -7.31 | *-57.94* | -1.27 | *-0.58* |
| **PBIAS** | ≤ 4 | -595.10 | *-335.27* | -6,359.73 | *-9,996.76* | -1,680.21 | *-889.04* |
| | ≤ 520 | -17.96 | *-25.19* | -987.01 | *-361.25* | -143.29 | *-32.72* |
| | ≤ 2,500 | 58.66 | *-5.18* | -402.14 | *-476.37* | -51.23 | *-9.59* |
| **r** | ≤ 4 | 0.24 | *0.38* | 0.47 | *0.35* | 0.39 | *0.54* |
| | ≤ 520 | 0.26 | *0.42* | 0.51 | *0.37* | 0.43 | *0.60* |
| | ≤ 2,500 | 0.26 | *0.47* | 0.51 | *0.41* | 0.45 | *0.66* |

| | | ORCHIDEE | PCR-GLOBWB | SURFEX-TRIP | W3RA | Ensemble mean |
|---|---|---|---|---|---|---|
| **NSE** | ≤ 4 | -4,301.50 | -176,842.51 | -2,938.32 | -35,536,946.62 | -5,645.16 |
| | ≤ 520 | -576.83 | -220.57 | -31.33 | -44,933.26 | -59.92 |
| | ≤ 2,500 | -1.30 | -8.80 | -0.54 | -1,878.46 | -1.17 |
| **PBIAS** | ≤ 4 | -7,049.09 | -8,661.80 | -2,415.21 | -235,229.53 | -5,108.67 |
| | ≤ 520 | -1,954.28 | -714.12 | -243.26 | -21,748.70 | -804.86 |
| | ≤ 2,500 | -103.64 | -91.09 | -57.74 | -5,014.81 | -188.17 |
| **r** | ≤ 4 | 0.32 | 0.12 | 0.45 | 0.31 | 0.46 |
| | ≤ 520 | 0.34 | 0.13 | 0.50 | 0.33 | 0.49 |
| | ≤ 2,500 | 0.31 | 0.13 | 0.52 | 0.36 | 0.49 |