# Peer review of "The potential of global re-analysis datasets in identifying flood events in Southern Africa"

_Hydrology and Earth System Sciences, 2018_

## Referee Comment (RC1) · Anonymous Referee #1 · 30 May 2018

General Comments

This paper assesses the potential of using re-analysis datasets with hydrological models to identify flood events in the Limpopo River basin. They evaluate climatological forcing's at 0.5 and 0.25 deg spatial resolution, and different hydrological and land surface models of the WRR datasets. While it is a model intercomparison paper, the objective is to identify timing and magnitude of floods. The novel aspect of the article is a flood detection comparison with reported observed flood damaged, which tries to link what is modeled to the actual impacts. The analysis focus on evaluating coarse spatiotemporal resolution dataset and models, which are not the most up to date and appropriated to assess local scale floods in small catchments. As the current generation of land surface and hydrological models are currently available at much higher

resolution (i.e., 5-10-5 km), these models could potentially be more appropriated and yield better skill in detecting floods. Nonetheless, I understand that the authors are constrained by the data and models available at the WRR dataset. However, WRR could have been updated for a more novel study. The paper is clear and concise, and the authors acknowledge limitations on data, models, and analysis, and well as listed aspects for improvement. Despite the limitations, this study intended to inform the scientific community on the potentials and limitation of currently available data and model for flood applications.

Specific Comments / Major Points

1. Page 2 Line 19 and Page 3 line 8: Can you expand the explanation of the term "spatially symmetrical"?

2. Page 3 Line 21: I would say . . . managing floods at the regional and basin scales . . .. I'm not sure to what extent forecasting at 0.25deg resolution is aiding flooding management at local scales.

3. Section 2.3.2 Disaster Data: I understand flood damage data is scarce and has its several limitations, which leads to data aggregation as an alternative to consolidate a standard analysis. However, it would be interesting to see few point results for maybe one or two cases where the location and time of the flood events are reported, and how do the models perform regarding flood timing, magnitude, and detection. This additional analysis would bring more meaningful insights on the potential use of these models for flooding management and flood detection, rather than a sub-basin aggregation.

4. Section 3.1: The models, in the context they were applied in this study, were not designed to evaluate discharge and floods at small catchments with $< 4km^2$, as the grid size is of at least $\sim$625 km$^2$ (0.25deg). As an (expected) result, the timing, magnitude, and flood detection are poorly captured. As these models are not appropriated to be applied in small catchments, can you expand on what is the purpose of evalu-
ating the small catchments in this study, why is it a reasonable approach, and which knowledge/information do you expect the scientific community will gain from it?

5. Section 3.1: Coefficient of determination could be used instead of linear Pearson to represent how much of the variability can be represented by the proposed models.

6. Page 11 Line 11. Can you expand on why do you think LISFLOOD perform worst using higher resolution forcing's data? I'd guess something related to model calibration.

7. This study was conducted considering data and models at daily time resolution, can you comment about the implications of temporal resolution on the forcing's data and modeling on the identification of short flashy floods. To what extent does it play a role in correctly identifying the flood category in places like southern Africa where rainfall if general driven by short and intense convective cells.

Technical Corrections / Minor Points

1. Page 2 Line 32: . . . determine climatic extremes as well as its uncertainties at global. . .

2. Page 3 Line 20: as an illustration, I'd list some examples of currently available flooding forecast systems currently available.

3. Page 4 Line 15: it would be nice to see the location of dams and reservoirs mapped in Figure 1., as it expands our sense about the basin dynamics and importance of representation of lake and reservoirs in hydrological models.

4. Page 14 Line 22: . . .whereas flood events occur at the basin or finer scales. . .

5. Page 15 Line 4. I'd say there is a critical need for both higher resolution re-analysis supporting data and flood forecasting systems to properly capture timing, intensity, and location of flood impacts.

6. Page 15 Line 22. I would change to something like: "This shows that some large-scale models (i.e., WaterGAP3) have some skill in capturing observed and reported

damaging floods, while others perform poorly. The finds here presented, highlights the importance of model intercomparison and evaluation studies to inform the scientific community on model's strengths and weakness as well as plausible applications".

7. Figure 1: Use a different color for the basin delineation; include the other river tributaries of a lower order, especially the ones where gauges were evaluated; use a different color for the circle and square.

8. Figure 2: Maybe you can check if a log scale in the y-axis of the NSE plots would improve the representation of the values around zero.

9. Figure 4: The 'X' points are very confusing and hard to follow through, maybe consider dots with a light line connecting them in the background.

10. Figure 5: Full name of POD and FAP in the figure caption.

11. Figure 6: Improve the figure quality regarding dpi. Change color scheme to opposite colors (i.e., red and blue) rather than light and dark (i.e., light blue and dark blue). Otherwise it's hard to see the differences.

---

## Referee Comment (RC2) · Anonymous Referee #2 · 20 Jun 2018

This study aims at evaluating the performances of several hydrological models in identifying flood events in South Africa. The models considered are members of the Water Resource Reanalysis (WRR) developed in the European Earth2Observe project. Models performances are evaluated using a frequency analysis and several skill scores related to flood detection. The authors also provide an interesting comparison with damaging events reported in three different disaster databases. Results convincingly show the ability of such models to capture the majority of flood events, despite their coarse resolution. Performances vary from one model to another, due to differences in model structure. The authors also pointed out the improvement due to the increase in spatial resolution (from 0.5° in WRR1 to 0.25° in WRR2). Before concluding, the authors discuss the main limitations of the method. The conclusions are consistent

with results presented all along the manuscript. The paper is well written and organized. This is a really interesting study and I think that the manuscript would be ready for publication provided that the authors address the following comments.

Major comment:

My major comment concerns the spatial resolution of the models. First, the authors often attribute the improvement in flood detection to the increases in spatial resolution. But in this case, the improvement can be due to three factors: (1) the improvement of models forcings: WFDEI (used in WRR1) and MSWEP (used in WRR2) rely on different methodologies (2) new model developments: each modeller involved in E2O included new developments in the models (e.g. multi-layer snow scheme in HTESSEL-CaMa, groundwater abstraction in LISFLOOD, reservoirs and water withdrawals in PCR-GLOBWB, aquifers and floodplains in SURFEX-CTRIP, etc.) (3) meteorological and hydrological processes better represented at higher resolution Many studies showed that simulated discharges are highly sensitive to meteorological forcing, especially precipitations, which are supposed to be of better quality in WRR2. Although points (1) and (3) are briefly mentioned in the discussion section (P13L2-5), I think they should be mentioned in the section presenting the models (section 2.2). Also all the conclusions on the differences in WRR1 and WRR2 performances should be put in this context. To have an idea of the impact of points (1) and (2) (without point (3)), the authors could consider the WRR2 version of the SURFEX-TRIP model which used the improved forcing (MSWEP) and new model developments but a spatial resolution of 0.5° for the routing scheme.

Another problem related to the spatial resolution is the selection of gauge stations (section 2.3.1). The authors mention that "the stations have upstream catchment areas that vary between 4 and 342,000 km2)". Is it realistic to compare observed and simulated discharges at stations with drainage area that small? Given that a model pixel has an area of approximately 2500 km2 for WRR1 and 650 km2 for WRR2, rivers with drainage area smaller than these thresholds are generally not represented in the models and the

correspondence between stations and model pixels is necessary wrong. This is consistent with authors results (P8L25-27, P12l30-32). This remark is mentioned P9L3-5, and in my opinion, stations with small drainage area should be excluded, even though there would remain a small number of stations. Also, could the authors give some details on the method used to select the model grid cell corresponding to each station? Was the river network of each model used to associate each station to a model grid cell?

Minor comments:

P3L31: "one of the largest basins"

P4L15-16: Have the impacts of such modifications on floods been studied already?

P5L11: Please provide a reference for "the Kolmogorov Smirnoff statistic for the Gumbel Extreme Value Distribution".

P5L21: Please add a few words to explain what the criteria from NatCatSERVICE is.

P6L19: In my understanding, hydrological extremes include floods but also droughts. This study only focuses on floods. The authors should carefully revise the use of "extremes" throughout the manuscript (including the following subtitle).

P7L12: "... for both the observed and the modelled discharges..."

P8L14: The areas mentioned in L13 are mainly related to the models spatial resolution. This should be specified.

P8L25-27: My guess is that the difference between WRR1 and WRR2 performances mostly comes from the forcings (rather than from the resolution).

P8L27-29: This result is hardly visible from Figure 2. It is more evident from Table 4a.

P9L8-11: I think this kind of conclusion should be built on larger basins only. Also, a fair comparison would only consider WRR1 so that the influence of forcings are excluded.

P10L9: "... were established using the observed climatology."

P10L26-27: These differences in performance are also (mainly?) the result of the improved forcings.

P12L5: "... and was evaluated by means of commonly used error statistics..."

P12L17-24: Would it be possible to get the dates of construction of major dams or reservoirs? It could be interesting to look at models performances before and after these dates.

P12L30: Improvements are also due to forcings improvements and new model development.

Figure 1: Please remove the marks within the inlet map. Also, please explain what the square and the circle represent (something like "stations used to illustrate the flood frequency analysis in section 3.2.1").

Figure 2: Would the NSE results be better presented using a log scale?

Figure 3, in the caption: "The index value refer to models with 0.5 degree resolution (MM1 and MO1) and 0.25 degree resolution (MM2 and MO2)."

Figure 6: The figure is not clear (small points/lines, close colours). Would results be better presented using box plots with mean, median, 1st and 4th quartiles?

Table 2: What are the impacts of changing theses values on the results?

Table 4, in the caption: Change "Figure 3, upper/lower panel" to "Table 4, upper/lower panel". Also, it seems that the selection of stations in each line of the table relies on a lower threshold, so the "lower or equal" sign should be "upper or equal".

[Figure]

---

## Author Comment (AC1) · 11 Jul 2018

**Response to reviewers**
**Manuscript for Hydrology and Earth System Sciences**
**Manuscript number: HESS-2018-164**
**Title: The potential of global re-analysis datasets in identifying flood events in Southern Africa**
**Authors: Gründemann, G.J., Werner, M., Veldkamp, T.I.E.**

**Referee #1:**

**General Comments**
**This paper assesses the potential of using re-analysis datasets with hydrological models to identify flood events in the Limpopo River basin. They evaluate climatological forcing's at 0.5 and 0.25 deg spatial resolution, and different hydrological and land surface models of the WRR datasets. While it is a model intercomparison paper, the objective is to identify timing and magnitude of floods. The novel aspect of the article is a flood detection comparison with reported observed flood damaged, which tries to link what is modeled to the actual impacts. The analysis focus on evaluating coarse spatiotemporal resolution dataset and models, which are not the most up to date and appropriated to assess local scale floods in small catchments. As the current generation of land surface and hydrological models are currently available at much higher resolution (i.e., 5-10-5 km), these models could potentially be more appropriated and yield better skill in detecting floods. Nonetheless, I understand that the authors are constrained by the data and models available at the WRR dataset. However, WRR could have been updated for a more novel study. The paper is clear and concise, and the authors acknowledge limitations on data, models, and analysis, and well as listed aspects for improvement. Despite the limitations, this study intended to inform the scientific community on the potentials and limitation of currently available data and model for flood applications.**
We thank the referee for taking the time to review our manuscript thoroughly and for his/her constructive comments. We are pleased that the referee values our work and is generally positive. The referee provided helpful comments in order to improve our manuscript. We have considered each of the comments carefully, which we will address here in detail. For the ease of reading we have copied the referee comments (in bold), and respond to each of the comments below.

**Specific comments**
**1. Page 2 Line 19 and Page 3 line 8: Can you expand the explanation of the term "spatially symmetrical"?**
We thank the referee for pointing out that this was not clear. By observational data being "spatially symmetrical" we mean that the data is evenly distributed across the study area. In the case of the Limpopo River Basin most of the available data as well as resources (financial, institutional) are located in South Africa, thus the data is not spatially symmetrical, which is mentioned in the next sentence (page 2 lines 19-20). We will change this in our revised manuscript to "not evenly distributed across the riparian countries, with most gauges in South Africa".

**2. Page 3 Line 21: I would say … managing floods at the regional and basin scales …. I'm not sure to what extent forecasting at 0.25deg resolution is aiding flooding management at local scales.**
We thank the referee for the comment. As the scope of our paper is to assess the scale up to which global models are able to provide useful information for small-scale flood risk management in areas with insufficient observational data, we used the word "local". As this was not completely clear, we will modify the text to "… managing floods at the regional and sub-basin scales.".

**3. Section 2.3.2 Disaster Data: I understand flood damage data is scarce and has its several limitations, which leads to data aggregation as an alternative to consolidate a standard analysis.**

**However, it would be interesting to see few point results for maybe one or two cases where the location and time of the flood events are reported, and how do the models perform regarding flood timing, magnitude, and detection. This additional analysis would bring more meaningful insights on the potential use of these models for flooding management and flood detection, rather than a sub-basin aggregation.**

Indeed, we agree that it indeed would be highly interesting. We will investigate the possibilities for adding this aspect in our revised manuscript.

**4. Section 3.1: The models, in the context they were applied in this study, were not designed to evaluate discharge and floods at small catchments with < 4km2, as the grid size is of at least _625 km2 (0.25deg). As an (expected) result, the timing, magnitude, and flood detection are poorly captured. As these models are not appropriated to be applied in small catchments, can you expand on what is the purpose of evaluating the small catchments in this study, why is it a reasonable approach, and which knowledge/information do you expect the scientific community will gain from it?**

Many thanks for this thought-provoking point. The reason we have decided to include the smallest sub-catchments as well, is because part of the scope of our paper is to determine the area up to which the models are still able to represent the hydrological behaviour. As shown in the results presented in Section 3.1 (page 8 line 25), the models were shown to capture the hydrology of the river for sub-catchments that are larger on the order of 520 km$^2$ for WRR2, which is on the order of the 625 km$^2$ of the cells size of the 0.25 degree models . Including the smaller catchments provides information about the scale up to which we can use such models that are included in the WRR dataset. This has, to our knowledge, not been studied before it is valuable for the scientific research community. Particularly since there are many regions that have to rely on such global data as the observational data is insufficient for localized models. Furthermore, as is for instance shown by Figure 3 where we analyse the Spookspruit river, with a sub-catchment area of only 252 km$^2$, global-scale models do actually have skill in capturing the variability (indicated by MM1 and MM2), even though the actual values are indeed overestimated. We will, however, rephrase this in our revised manuscript as this was not entirely clear. Since both referees raised this point, we are nevertheless willing to disregard the smallest stations from our analysis. We are therefore interested in the opinion of the editor regarding this issue. We also consider of scientific interest that for both the coarser and the finer resolution models, the threshold is on the order of the cell size. This holds promise for the continuing effort of modelling research groups in developing increased resolution (global models).

**5. Section 3.1: Coefficient of determination could be used instead of linear Pearson to represent how much of the variability can be represented by the proposed models.**

Indeed, we agree that the coefficient of determination could have also been used, as they both describe the degree of collinearity between the modelled and observed data, though in a different way. The coefficient of determination describes the proportion of the variance in the measured data explained by the model, whereas Pearson's correlation coefficient is an index of the degree of linear relationship between observed and simulated data. We were interested in this linear relationship, in addition to the NSE and PBIAS, which is why we chose for Pearson's correlation coefficient above the coefficient of determination.

**6. Page 11 Line 11. Can you expand on why do you think LISFLOOD perform worst using higher resolution forcing's data? I'd guess something related to model calibration.**

We thank the referee for raising this interesting point. We have greatly looked into the reason why LISFLOOD could be performing worse in a higher resolution. There are a number of factors that could contribute to this. First of all, the models in WRR2 had further modifications in respect to WRR1, apart from the different forcing and higher resolution. For LISFLOOD the modifications include an increased number of lakes and reservoirs, improvement of irrigation water demand and groundwater abstraction and an increased number of soil layers (Arduini et al., 2017; Dutra et al., 2017). This could be part of

the answer as to why the improved forcing and higher spatial resolution did not result in an improved performance. Secondly, we agree with the referee that it could indeed also be related to the model calibration. LISFLOOD was somewhat calibrated for WRR1 using eleven parameters for 24 large catchments (Dutra et al., 2015), but LISFLOOD was not calibrated for WRR2 (Dutra et al., 2017). Instead, the parameterisation of WRR1 was also used for WRR2, even though the alterations to the model require an updated calibration (Arduini et al., 2017; Dutra et al., 2017). We will include the reasons for the worse performance of LISFLOOD in WRR2 in our revised manuscript.

**7. This study was conducted considering data and models at daily time resolution, can you comment about the implications of temporal resolution on the forcing's data and modeling on the identification of short flashy floods. To what extent does it play a role in correctly identifying the flood category in places like southern Africa where rainfall if general driven by short and intense convective cells.**

The forcing data of MSWEP is actually available at a 3-hourly resolution. However, as most models operate at the daily timestep (in the WRR dataset LISFLOOD, PCR-GLOBWB, W3RA and WaterGAP3) and since the output is also at the daily step, the forcing data with a sub-daily resolution is generally not used. Furthermore, sub-daily data is generally not available in these areas. Some of the rain- and discharge gauges report hourly values, but most only at the daily timestep. In order to analyse the short flash floods, even shorter reporting time-steps (every minute or 5-minutes) would be preferred. Apart from the high resolution data a very high resolution local model is also needed to model the short flash floods (López et al., 2016). For these models, the forcing data used in the WRR dataset would be insufficient. Our scope, however, is to look at the potential of available datasets at the global scale, and not to model at the high resolution the occurrence of flash floods. Indeed, these flash floods are likely not well captured in the data, and most research does not take this kind of flood events into consideration. That is the exact reason why we are also interested in including this in our research.

**Technical Corrections / Minor Points**
**1. Page 2 Line 32: … determine climatic extremes as well as its uncertainties at global…**
We thank the referee for pointing this out, the sentence will be altered.

**2. Page 3 Line 20: as an illustration, I'd list some examples of currently available flooding forecast systems currently available.**
The suggestion is highly appreciated, we will look into it.

**3. Page 4 Line 15: it would be nice to see the location of dams and reservoirs mapped in Figure 1., as it expands our sense about the basin dynamics and importance of representation of lake and reservoirs in hydrological models.**
We agree with this comment, we will add this in the revised version of our paper.

**4. Page 14 Line 22: …whereas flood events occur at the basin or finer scales…**
We assume the referee is referring to Page 13 Line 22: "whereas flood events occur at the basin scale". In this case, we agree with the referee and will modify this sentence text accordingly.

**5. Page 15 Line 4. I'd say there is a critical need for both higher resolution re-analysis supporting data and flood forecasting systems to properly capture timing, intensity, and location of flood impacts.**
We agree with the referee that there is a critical need for both the re-analysis data as well as flood forecasting systems. Such global models as used in this re-analysis are also employed in global forecasting systems, such as GloFas (Alfieri et al., 2013). GloFas uses the HTESSEL model which is also considered here, but then at a resolution of 0.1 degrees. We will include this in our revised manuscript.

**6. Page 15 Line 22. I would change to something like: "This shows that some largescale models (i.e., WaterGAP3) have some skill in capturing observed and reported damaging floods, while others**

**perform poorly. The finds here presented, highlights the importance of model intercomparison and evaluation studies to inform the scientific community on model's strengths and weakness as well as plausible applications".**

Many thanks for this comment, however, we do not fully agree here. In this particular study that focussed on the Limpopo basin WaterGAP3 performed better, but it is not sure if this is a general conclusion. There are many instances possible where other models would perform better. For instance, if this study would be repeated elsewhere, or if the discharge outputs of other models would be compared, or if the models would have had another forcing, or if all models would have been calibrated.

**7. Figure 1: Use a different color for the basin delineation; include the other river tributaries of a lower order, especially the ones where gauges were evaluated; use a different color for the circle and square.**

We thank the author for the suggestions to improve this figure, we will alter the figure where possible to include these points, as well as add the dams and/or reservoirs (as was pointed out in the Technical Corrections / Minor Points 3).

**8. Figure 2: Maybe you can check if a log scale in the y-axis of the NSE plots would improve the representation of the values around zero.**

Thank you for your comment, which was also raised by the second referee. We will take a closer look at improving the graphs for the NSE in this figure.

[Figure]

**9. Figure 4: The 'X' points are very confusing and hard to follow through, maybe consider dots with a light line connecting them in the background.**

Thank you, we have modified this figure, see below for WRR1. This indeed makes the figure clearer.

[Figure]

**10. Figure 5: Full name of POD and FAR in the figure caption.**
Thank you for pointing this out, we will include the full name in the paper.

**11. Figure 6: Improve the figure quality regarding dpi. Change color scheme to opposite colors (i.e., red and blue) rather than light and dark (i.e., light blue and dark blue). Otherwise it's hard to see the differences.**
Thank you for raising this issue, which was also mentioned by the second referee. We will change the figure in the revised version of the paper.

**References used in this revision response:**
Alfieri, L., Burek, P., Dutra, E., Krzeminski, B., Muraro, D., Thielen, J. and Pappenberger, F.: GloFAS-global ensemble streamflow forecasting and flood early warning, Hydrol. Earth Syst. Sci., 17(3), 1161–1175, doi:10.5194/hess-17-1161-2013, 2013.

Arduini, G., Fink, G., Martinez de la Torre, A., Nikolopoulos, E., Anagnostou, E., Balsamo, G. and Boussetta, S.: End-user-focused improvements and descriptions of the advances introduced between the WRR tier1 and WRR tier2., 2017.

Dutra, E., Balsamo, G., Calvet, J.-C., Minvielle, M., Eisner, S., Fink, G., Pessenteiner, S., Orth, R., Burke, S., van Dijk, A. I. J. M., Polcher, J., Beck, H. E. and Martinez de la Torre, A.: Report on the current state-of-the-art Water Resources Reanalysis., 2015.

Dutra, E., Balsamo, G., Calvet, J.-C., Munier, S., Burke, S., Fink, G., van Dijk, A. I. J. M., Martinez de la Torre, A., van Beek, R., de Roo, A. and Polcher, J.: Report on the improved Water Resources Reanalysis (WRR2)., 2017.

Huh, S., Dickey, D. A., Meador, M. R. and Ruhl, K. E.: Temporal analysis of the frequency and duration of low and high streamflow : years of record needed to characterize streamflow variability, J. Hydrol., 78–94, doi:10.1016/j.jhydrol.2004.12.008, 2005.

López, P.L., Wanders, N., Schellekens, J., Renzullo, L. J. and Sutanudjaja, E. H.: Improved large-scale hydrological modelling through the assimilation of streamflow and downscaled satellite soil moisture observations, Hydrol. Earth Syst. Sci., 20, 3059–3076, doi:10.5194/hess-20-3059-2016, 2016.

---

## Author Comment (AC2) · 11 Jul 2018

Response to reviewers
Manuscript for Hydrology and Earth System Sciences
Manuscript number: HESS-2018-164
Title: The potential of global re-analysis datasets in identifying flood events in Southern Africa
Authors: Gründemann, G.J., Werner, M., Veldkamp, T.I.E.

**Referee #2:**

**This study aims at evaluating the performances of several hydrological models in identifying flood events in South Africa. The models considered are members of the Water Resource Reanalysis (WRR) developed in the European Earth2Observe project. Models performances are evaluated using a frequency analysis and several skill scores related to flood detection. The authors also provide an interesting comparison with damaging events reported in three different disaster databases. Results convincingly show the ability of such models to capture the majority of flood events, despite their coarse resolution. Performances vary from one model to another, due to differences in model structure. The authors also pointed out the improvement due to the increase in spatial resolution (from 0.5_ in WRR1 to 0.25_ in WRR2). Before concluding, the authors discuss the main limitations of the method. The conclusions are consistent with results presented all along the manuscript. The paper is well written and organized. This is a really interesting study and I think that the manuscript would be ready for publication provided that the authors address the following comments.**
We thank the anonymous referee for his/her time in carefully reviewing our manuscript. The referee provided us with helpful comments, which will improve our manuscript. We are glad that the referee finds our work of importance for the scientific community to consider publication in HESS after revisions. We have carefully studied each of the remarks and will address them here in detail. We have copied the referee comments (in bold), and respond to each them below.

**Major comment:**
**My major comment concerns the spatial resolution of the models. First, the authors often attribute the improvement in flood detection to the increases in spatial resolution. But in this case, the improvement can be due to three factors: (1) the improvement of models forcings: WFDEI (used in WRR1) and MSWEP (used in WRR2) rely on different methodologies (2) new model developments: each modeller involved in E2O included new developments in the models (e.g. multi-layer snow scheme in HTESSELCaMa, groundwater abstraction in LISFLOOD, reservoirs and water withdrawals in PCR-GLOBWB, aquifers and floodplains in SURFEX-CTRIP, etc.) (3) meteorological and hydrological processes better represented at higher resolution Many studies showed that simulated discharges are highly sensitive to meteorological forcing, especially precipitations, which are supposed to be of better quality in WRR2. Although points (1) and (3) are briefly mentioned in the discussion section (P13L2-5), I think they should be mentioned in the section presenting the models (section 2.2). Also all the conclusions on the differences in WRR1 and WRR2 performances should be put in this context. To have an idea of the impact of points (1) and (2) (without point (3)), the authors could consider the WRR2 version of the SURFEX-TRIP model which used the improved forcing (MSWEP) and new model developments but a spatial resolution of 0.5_ for the routing scheme.**
We thank the referee for raising this interesting comment. We were actually considering to include SURFEX-TRIP in our initial analysis, but decided against it due to possible inconsistency reasons. All model outputs in WRR2 are available at 0.25 degrees, except for the river discharge of the SURFEX-TRIP model as the same routing scheme used in WRR1 as applied. We therefore decided to focus solely on the models that did meet the standards agreed upon by EartH2Observe.

However, as this remark has been raised, we took a look at the differences in the error statistics of SURFEX-TRIP in WRR1 and WRR2, as can be seen in the table below that is part of Table 4 in our manuscript. As can be seen, there is an improvement of WRR2 compared to WRR1, and we thus have

decided to include SURFEX-TRIP WRR2 at 0.5 degrees in our revised manuscript. We will perform further analysis and based on this new information revise our manuscript, put the differences between WRR1 and WRR2 in the context proposed by the referee, and update Figures 2, 4, 5 and 6 specifically.

| | Stations with upstream catchment area [km$^2$] | SURFEX-TRIP | |
|---|---|---|---|
| | | wrr1 | wrr2 |
| NSE | ≥ 4 | -2,938.32 | -1,147.33 |
| | ≥ 520 | -31.33 | -21.73 |
| | ≥ 2,500 | -0.54 | -1.21 |
| PBIAS | ≥ 4 | -2,415.21 | -1,176.01 |
| | ≥ 520 | -243.26 | -167.95 |
| | ≥ 2,500 | -57.74 | -74.57 |
| r | ≥ 4 | 0.45 | 0.50 |
| | ≥ 520 | 0.50 | 0.56 |
| | ≥ 2,500 | 0.52 | 0.60 |

**Another problem related to the spatial resolution is the selection of gauge stations (section 2.3.1). The authors mention that "the stations have upstream catchment areas that vary between 4 and 342,000 km2)". Is it realistic to compare observed and simulated discharges at stations with drainage area that small? Given that a model pixel has an area of approximately 2500 km2 for WRR1 and 650 km2 for WRR2, rivers with drainage area smaller than these thresholds are generally not represented in the models and the correspondence between stations and model pixels is necessary wrong. This is consistent with authors results (P8L25-27, P12l30-32). This remark is mentioned P9L3-5, and in my opinion, stations with small drainage area should be excluded, even though there would remain a small number of stations. Also, could the authors give some details on the method used to select the model grid cell corresponding to each station? Was the river network of each model used to associate each station to a model grid cell?**

We thank the referee for this comment, which was also mentioned by the first referee. Our answer is similar and is copied below. The reason we have decided to include the smallest sub-catchments as well, is because part of the scope of our paper is to determine the area up to which the models are still able to represent the hydrological behaviour. As shown in the results presented in Section 3.1 (page 8 line 25), the models were shown to capture the hydrology of the river for sub-catchments that are larger on the order of 520 km$^2$ for WRR2, which is on the order of the 625 km$^2$ of the cells size of the 0.25 degree models . Including the smaller catchments provides information about the scale up to which we can use such models that are included in the WRR dataset. This has, to our knowledge, not been studied before it is valuable for the scientific research community. Particularly since there are many regions that have to rely on such global data as the observational data is insufficient for localized models. Furthermore, as is for instance shown by Figure 3 where we analyse the Spookspruit river, with a sub-catchment area of only 252 km$^2$, global-scale models do actually have skill in capturing the variability (indicated by MM1 and MM2), even though the actual values are indeed overestimated. We will, however, rephrase this in our revised manuscript as this was not entirely clear. Since both referees raised this point, we are nevertheless willing to disregard the smallest stations from our analysis. We are therefore interested in the opinion of the editor regarding this issue. We also consider of scientific interest that for both the coarser and the finer resolution models, the threshold is on the order of the cell size. This holds promise for the continuing effort of modelling research groups in developing increased resolution (global models).

Concerning the method we used to select the model-grid cell for each station, we looked at the location of the model and selected the exact model-grid cell that it was located in. Unfortunately we did not have any access to the river network of each model, so we couldn't use this to associate each station to the exact model-grid cell. We did, however, check the eight surrounding cells and select the cell with the highest discharge as the river cell corresponding to the discharge gauge. This did work for larger basins, but it did not for the smaller ones if there was a large river in a neighbouring cell.

**Minor comments:**
**P3L31: "one of the largest basins"**
Thank you, we will modify this sentence.

**P4L15-16: Have the impacts of such modifications on floods been studied already?**
Thank you, there has not yet, to our knowledge, been a study into the impacts of these types of human modifications in the Limpopo Basin in terms of floods. Previous studies have, however, looked into the impacts of human modifications such as dams in other rivers, such as the study by Fitzhugh and Vogel (2011) for the United States specifically.

**P5L11: Please provide a reference for "the Kolmogorov Smirnoff statistic for the Gumbel Extreme Value Distribution".**
The reference will be provided.

**P5L21: Please add a few words to explain what the criteria from NatCatSERVICE is.**
Thank you for your comment. In order to determine the severity of a disaster event NatCatSERVICE uses the number of fatalities and overall losses as criteria. We will add this in our paper.

**P6L19: In my understanding, hydrological extremes include floods but also droughts. This study only focuses on floods. The authors should carefully revise the use of "extremes" throughout the manuscript (including the following subtitle).**
We thank the referee for pointing this out. Indeed we have used the term "extremes" mainly regarding floods in our paper. We will revise our use of this term throughout the entire paper.

**P7L12: "…for both the observed and the modelled discharges…"**
We will change this in the final version of our paper.

**P8L14: The areas mentioned in L13 are mainly related to the models spatial resolution. This should be specified.**
Thank you for pointing out that this was not clear, we will clarify this further.

**P8L25-27: My guess is that the difference between WRR1 and WRR2 performances mostly comes from the forcings (rather than from the resolution).**
We thank the referee for this interesting comment. We have taken a closer look at the causes of the difference in WRR1 and WRR2. Indeed the different forcing used in WRR2 could be one of the reasons. However, model improvements and the improved resolution are likely to contribute to the improved performance. That model structure and resolution does influence the results can be seen by how improved forcing leads to differing changes to model performance, depending on the model. To unravel the influence improved forcing and model improvements, the higher resolution models could be re-run with the (resampled) low resolution forcing data. While this will reveal no doubt interesting results we consider this as outside the scope of the current paper.

**P8L27-29: This result is hardly visible from Figure 2. It is more evident from Table 4a.**
Thank you for pointing this out, we will include explicit reference to the table.

**P9L8-11: I think this kind of conclusion should be built on larger basins only. Also, a fair comparison would only consider WRR1 so that the influence of forcings are excluded.**
Thank you for raising this issue. We shall base this conclusion solely on the largest stations. Additionally, we actually did only consider the models in WRR1 for this conclusion, but we apologise if that was not clear. We clarify this in the revised version of our paper.

**P10L9: "…were established using the observed climatology."**
Thank you, we will modify this sentence.

**P10L26-27: These differences in performance are also (mainly?) the result of the improved forcings.**
Same as for the first major comment, we will revise this.

**P12L5: "…and was evaluated by means of commonly used error statistics..."**
Thank you , we will alter this.

**P12L17-24: Would it be possible to get the dates of construction of major dams or reservoirs? It could be interesting to look at models performances before and after these dates.**
We agree with the referee that this would indeed be interesting. In order to analyse the hydrologic impacts of dams discharge statistics from periods before and after the construction of the dam need to be analysed. It is recommended to have 20 years of data both pre- as well as post construction of the dam (Huh et al., 2005). As the WRR dataset is available between 1980 and 2012, this would mean that at maximum 16 years of data would be available, if the dam was constructed in the middle of the period of record (1996). We have analysed data on dams in the basin and the dates of commissioning. From the available information we found that o major dam was completed in 1996 or the three years before and after. The closest major dam completions are 1987 (Flag Boshielo Dam in South Africa) and 2000 (Letsibogo Dam in Botswana). We therefore think that there is insufficient data to do a robust trend-analysis. Furthermore, not all models include reservoirs, and those that do, do not do so in the same manner. Some models have "static" reservoirs, whereas others include the reservoirs over time. Therefore, doing an integrated analysis for all models equally is beyond the scope of our paper, and would be a separate study.

**P12L30: Improvements are also due to forcings improvements and new model development.**
Same as for the first major comment, we will revise this.

**Figure 1: Please remove the marks within the inlet map. Also, please explain what the square and the circle represent (something like "stations used to illustrate the flood frequency analysis in section 3.2.1").**
Thank you, we will remove the marks and add a further explanation regarding the two stations.

**Figure 2: Would the NSE results be better presented using a log scale?**
Thank you for your comment, which was also raised by the other referee. We will alter the figure and make the NSE results more clear.

**Figure 3, in the caption: "The index value refer to models with 0.5 degree resolution (MM1 and MO1) and 0.25 degree resolution (MM2 and MO2)."**
Thank you, we will change the caption in Figure 3.

**Figure 6: The figure is not clear (small points/lines, close colours). Would results be better presented using box plots with mean, median, 1st and 4th quartiles?**
Thank you for pointing this out. As the first referee also pointed this out, we feel like our research will benefit from this and we will hence modify this figure.

**Table 2: What are the impacts of changing theses values on the results?**
Changing these values is not significantly impacting the results of this research. The only instance where these thresholds are used is for constructing Figure 3. This figure is merely an illustration of two example gauges to illustrate the difference between the model climatology versus the observed climatology. If higher thresholds would be chosen, they would not be exceeded as much, whereas if lower thresholds would be chosen, they would be exceeded more often.

**Table 4, in the caption: Change "Figure 3, upper/lower panel" to "Table 4, upper/lower panel". Also, it seems that the selection of stations in each line of the table relies on a lower threshold, so the "lower or equal" sign should be "upper or equal".**
Thank you for pointing this out, the caption and "lower or equal" signs were incorrect. We will modify the table and caption accordingly.

**References used in this revision response:**
Fitzhugh, T.W. and Vogel, R.M.: The impact of dams on flood flows in the United States, River Res. Appl., 310, 1192-1215, doi:10.1002/rra, 2011.